# Differential Allelic Expression among Long Non-Coding RNAs

**DOI:** 10.3390/ncrna7040066

**Published:** 2021-10-22

**Authors:** Michael B. Heskett, Paul T. Spellman, Mathew J. Thayer

**Affiliations:** 1Department of Genetics, Oregon Health & Science University, Portland, OR 97239, USA; heskett@ohsu.edu (M.B.H.); spellmap@ohsu.edu (P.T.S.); 2Department of Chemical Physiology and Biochemistry, Oregon Health & Science University, Portland, OR 97239, USA

**Keywords:** lncRNA, allele specific, ASAR, Xist, epigenetics

## Abstract

Long non-coding RNAs (lncRNA) comprise a diverse group of non-protein-coding RNAs >200 bp in length that are involved in various normal cellular processes and disease states, and can affect coding gene expression through mechanisms in *cis* or in *trans*. Since the discovery of the first functional lncRNAs transcribed by RNA Polymerase II, H19 and Xist, many others have been identified and noted for their unusual transcriptional pattern, whereby expression from one chromosome homolog is strongly favored over the other, also known as mono-allelic or differential allelic expression. lncRNAs with differential allelic expression have been observed to play critical roles in developmental gene regulation, chromosome structure, and disease. Here, we will focus on known examples of differential allelic expression of lncRNAs and highlight recent research describing functional lncRNAs expressed from both imprinted and random mono-allelic expression domains.

## 1. Introduction

Sequencing of the human genome and other eukaryotic species from both close and distant lineages has revealed that the number and composition of protein-coding genes are unmistakably similar, fueling the idea that development of complex organisms may be in large part due to non-coding elements. While ~1% of the genome codes for protein, RNA extracted from multiple tissues and subcellular fractions reproducibly covers ~62% of genomic bases [1], demonstrating that a large portion of the genome is transcribed at some point, but remains of unknown function. Characterization of the non-coding portion of the genome has yielded a growing list of long non-coding RNAs (lncRNAs), defined as >200 nt transcript with no perceivable coding potential [2]. While RNA Pol I and RNA Pol III are responsible for transcription of the majority of the ncRNA mass in higher eukaryotes (i.e., tRNAs and rRNAs), this review is focused on the lncRNAs that are transcribed by RNA Pol II. Canonical RNA Pol II lncRNAs have exon/intron organization that is similar to protein-coding mRNAs, suggesting a shared evolutionary origin [2]. Although a comprehensive description of lncRNA structure and function has not yet been rendered, the observation that one third of all known lncRNAs are primate specific, including many human-specific transcripts [2,3], as well as the highly tissue-specific expression patterns of lncRNAs [4], suggests importance in development of the complex organ systems of higher eukaryotes. lncRNAs affect genomic phenotypes through a variety of mechanisms in *cis* or in *trans,* including blocking or enhancing activity of chromatin modifying proteins, directly binding DNA [5], and regulating gene expression through lncRNA transcript-dependent or lncRNA transcription-dependent mechanisms (reviewed in: [6]). Here, we focus on the subset of lncRNAs that are expressed preferentially from one chromosome homolog, a phenomenon known as mono-allelic, or differential allelic expression (DAE).

In the diploid human genome, it is currently accepted that most genes are transcribed simultaneously from both homologs, known as bi-allelic expression [7]. However, genetic and epigenetic factors can result in DAE. For example, expression quantitative trait loci (eQTLs), are genetic polymorphisms spread throughout the genome that result in DAE when present as heterozygotes [8,9]. In contrast, DAE can also occur via epigenetic mechanisms that have been well studied in the context of antigen receptors in the immune system [10,11], the odorant receptors in the olfactory system [12], and during X chromosome inactivation in female mammals [13]. In the immune and olfactory systems, DAE is utilized to generate vast diversity of antigen and odorant recognition. In the adaptive immune response, T and B cells undergo somatic recombination to exclude multiple alleles of polygenic loci, and cells that successfully express a single rearranged T-cell receptor or immunoglobulin gene (Ig) are selected for. A similar allelic exclusion process occurs in the olfactory system, and in the absence of somatic recombination results in each olfactory neuron expressing a single gene from a multigene family [12]. Genomic imprinting, mediated by stably inherited differential epigenetic modifications in the germline and somatic cells, is critical for development in human and mouse (Figure 1A,B). For example, at the imprinted region 15q11–13, known as the Prader–Willi/Angelman locus, disruption of paternal-specific expression of SNRPN and NDN, or disruption of maternal-specific expression of adjacent UBE3A, leads to severe developmental dysfunction [14]. In contrast to imprinting, the polarity of DAE may display an expression pattern that is selected at random between paternal and maternal homologs within cells of an individual. Within the context of dosage compensation on the X chromosome in females, epigenetic differences in DAE results in a mosaic pattern of expression with half of cells expressing the maternal allele and the other half expressing the paternal allele, a pattern of expression referred to as random mono-allelic expression (RME; Figure 1C). RME is most notably observed on the X-chromosome of female mammalian cells, where one of the X chromosomes is epigenetically remodeled into a transcriptionally inactive state, leading to strong DAE and achieving dosage compensation [13]. In addition to RME occurring on the X chromosome, between 2 and 10% of autosomal genes have also been observed to display RME, however, many studies have been performed using the highly heterozygous genome of the F1-hybrid mouse model system [15,16], which, given the ~500,000 years of evolutionary difference between these mouse strains, may not provide findings generalizable to the human population.

Autosomal RME is thought to play an important role in brain development through the process of neuronal individuality generated by functional heterozygosity [17]. Diverse and complex neural networks rely on selective responses of individual neurons, likely reflected by variation in surface receptor expression and axon guidance. Given that inherent genomic heterozygosity results in slight differences between each allele of every gene, RME of even a small number of autosomal genes can generate an extremely high number of unique allelic combinations (Figure 1D). One such system involves the clustered protocadherins (Pcdhs), cell adhesion proteins expressed in the mammalian brain that display RME and form heteromultimeric *cis*-tetramers, allowing for a combinatorial “explosion” of the number of possible unique neuronal identities [18]. The biological repercussions of RME are less clear outside of the context of dosage compensation and the immune, olfactory, and nervous systems.

## 2. Differential Allelic Expression of lncRNA by Genomic Imprinting

Of the few lncRNAs whose functions have been elucidated, many examples display DAE. The H19 lncRNA was the first RNA Pol II derived lncRNA discovered in the mammalian genome and was shortly thereafter found to display DAE via genomic imprinting [19,20]. Since then, many other imprinted lncRNAs including Airn, Kcnq1ot1, Snhg14, Gtl2, and Nespas have been described [21]. H19 is located downstream of the growth-promoting Igf2 gene and is expressed from the maternal allele. Maternal expression leads to interference with an Igf2 enhancer element in a process that is also dependent on CTCF, resulting in repressed maternal Igf2 expression [19]. On the paternal allele, the H19 promoter is methylated and silenced which allows Igf2 to interact with the distal enhancer [22]. At a separate imprinted locus, the Igf2r gene is also regulated by a lncRNA called Airn (or Air). Airn is located within an intron of the Igf2r gene and is expressed in an antisense manner on the paternal allele, disrupting paternal Igf2r coding gene expression. On the maternal allele, Airn is silenced by methylation, allowing for expression of the maternal Igf2r gene [22]. H19 is being investigated in the context of breast cancer [23,24], and was found to be overexpressed in gastric cancers and correlated with survival [25]. Another imprinted lncRNA, Kcnq1ot1, may function through a similar mechanism and is maternally imprinted, paternally expressed, and silences the paternal expression of 8–10 adjacent coding gene on chromosome 11 [21]. Kcnq1ot1 has been recently implicated in mediating pyroptosis in diabetic cardiomyopathy [26], and is implicated in multiple cancers via an miRNA sponge mechanism [27]. Imprinted lncRNA silencing of downstream genes can be explained by transcriptional interference; however, repression of upstream genes cannot be accounted for by this model, suggesting an additional mechanism is in play.

H19, Kcnq1ot1, Nespas, Snhg14, Gtl2, and others are all located within imprinted regions where antisense lncRNAs are involved in *cis*-regulation that may be tissue specific or occur at specific stages during development [21]. Although lncRNA-mediated, parent-of-origin-specific expression has been observed throughout the genome, the impact is still unclear. Parent-of-origin effects (POEs) [28] are not well understood, but initial studies have shown that maternally and paternally derived alleles, including significant examples in and near lncRNAs, can be differentially associated with quantitative phenotypes including low density lipoprotein cholesterol (LDL-C) and triglyceride levels, as well as body mass index [29]. For example, the lncRNA gene LINC01081, a regulator of the FOXF1 coding gene that is expressed in a parent- and tissue-specific manner, contains a variant that has an opposite association with LDL-C depending on its parent of origin context [29].

## 3. Differential Allelic Expression of lncRNAs in X-Inactivation

LncRNA Xist, located at the X-inactivation center (XCI) and exhibiting an RME pattern, is expressed from the inactive X and is the major effector of X-inactivation in female mammalian cells [30]. Xist RNA is retained within the inactive X chromosome territory, spreads to nearby gene dense regions on the inactive X and recruits repressive epigenetic modulators such as PRC2, resulting in remodeling and transcriptional repression of hundreds of genes across most of the X chromosome [31]. Models of Xist RNA function have been recently refined using time resolved super-resolution microscopy, demonstrating a feedback mechanism that links rates of synthesis and degradation, aided by anchoring of Xist molecules to the nuclear matrix [32]. A “jump and coupling” model has been proposed, whereby Xist undergoes an expansion phase with radial enlargement of Xist RNA territories, followed by a steady state phase where Xist localizes in a more stochastic manner [32].

Xist functions in concurrence with other DAE lncRNAs, including being subject to regulation *in-cis* by the lncRNA Tsix. Tsix is in the antisense orientation to Xist and serves to downregulate Xist on the active X [33]. Additionally, a promoter element of the Linx lncRNA acts as a long-range silencer to influence the choice of X chromosome to be inactivated [34,35,36]. The X-linked lncRNA Firre was originally reported to be expressed from both X homologs, but having an allele-specific role on the inactive X by recruiting CTCF and cohesin to maintain perinucleolar positioning of the X-chromosome [37]. However, recent research in additional mouse cell lines has shown that Firre displays DAE and is transcribed primarily from the active X. Deletion of Firre from the active X leads to a loss of the canonical H3K27me3 signal and spatial regulation of the inactive X, demonstrating a *trans*-acting function [38]. The discovery of the exclusive *trans*-acting function of Firre poses the question of how a lncRNA may dissociate from the chromosome from which it was expressed to locate and recognize the opposite homolog. The lncRNA XACT is also expressed during the pre-implantation stage of human embryos and in naive pluripotent stem cells, where XACT RNA “coats” the active X-chromosome to antagonize XIST accumulation [22]. XACT is not well-conserved among mammals and is not present in the mouse genome, suggesting a primate-specific function.

## 4. Autosomal Differential Allelic Expression of Non-Canonical lncRNAs Is Required for Normal DNA Replication Timing and Chromosome Stability

Several recent studies have described RME of the autosomal lncRNA genes *ASAR6*, *ASAR6-141*, and *ASAR15*, which are located on chromosomes 6, 6, and 15, respectively. Each of the ASAR lncRNA genes are >200 kb and express lncRNAs that are distinct from canonical RNA Pol II lncRNAs due to their very long length, lack of splicing, and lack of polyadenylation [39,40,41]. ASARs belong to a less well-studied group of lncRNAs that have also been termed “very long intergenic” (vlincRNAs) or “macro” lncRNAs, and are thought to make up the majority of nuclear-encoded non-ribosomal RNA in the cell [42,43,44]. The DAE associated with ASAR lncRNAs is characterized by the presence of one large RNA fluorescent in situ hybridization (FISH) signal in individual cells that is retained within the chromosome territory from which the RNA is expressed [39,40,41,45]. RME of ASARs has been established by RNA/DNA FISH, RT-PCR following by sequencing at heterozygous SNPs, allele-specific RT-PCR, and RNAseq [39,40]. In contrast to imprinted expression, the polarity of DAE of ASARs is epigenetically distinct between single cell-derived clones of cells derived from the same individual [45]. While the choice of homolog for XIST expression and subsequent X-inactivation occurs early in embryonic development, it is currently unknown when and how DAE is established and maintained for ASARs. Of note, ASAR6 is required for the DAE of the adjacent protein-coding gene FUT9 and the lncRNA gene FHL5OST [45].

In addition to their unusual nuclear localization pattern and DAE, ASARs are essential for normal chromosome function. Deletion or disruption of the expressed alleles of the three known ASAR lncRNA genes leads to delayed DNA replication timing (DRT) of the entire chromosome, followed by a subsequent delay in mitotic chromosome condensation (DMC), and chromosome structure instability [46,47]. Similarly, disruption of Xist on the inactive X chromosome in mice results in DRT [48]. ASAR function has also been demonstrated using a gain of function assay. For example, ectopic integration of an *ASAR6* transgene into mouse chromosome 3 leads to the DRT/DMC phenotype, which can be subsequently rescued by antisense oligonucleotides that knock down the ASAR6 RNA [49], indicating that ASAR function is transcript dependent [49].

Utilizing genetic deletion or ectopic integration assays of a full-length L1PA2 transposable element (oriented in the antisense direction) from within *ASAR6* leads to the DRT phenotype, suggesting a critical function of the antisense strand of the L1PA2 in the ASAR6 RNA product in controlling replication timing [49]. In addition, ectopic integration of a transgene expressing the antisense strand of an L1PA13 from *ASAR15*, results in the DRT/DMC phenotype on mouse chromosomes. Thus, loss of function of ASARs on human chromosomes or ectopic integration of ASARs into mouse chromosomes leads to the same phenotype, suggesting a dominant interfering activity of the ASAR transgenes [49]. Because chromosomes with DRT/DMC are often seen in mitotic preparations of cancer cells, it is proposed that disruption of ASAR genes may be a widespread mechanism leading to chromosome structure instability in cancer [50]. One form of chromosome structure instability in cancer is known as chromothripsis, whereby tens to thousands of rearrangements occur on a single chromosome [51]. A plausible model of chromothripsis development currently supported by the field describes a single catastrophic chromosome breakage event occurring during mitosis, resulting in many chromosome fragments, followed by repair in a highly disjointed order and orientation [52]. Due to the frequency and close proximity of chromosome breakpoints in chromothripsis, improper mitotic condensation is suspected as a likely precursor to a chromothriptic event, warranting further investigation into a possible link between DRT/DMC resulting from ASAR disruption and chromothripsis.

The observation of abundant repeat-rich lncRNA associating with chromatin is not unique to ASARs, and has been previously described as a characteristic of Cot-1 RNA [53]. Cot-1 DNA represents sequences containing a high concentration of LINEs and SINEs that have been traditionally used to block non-specific binding of repeat rich DNA to hybridization probes. Cot-1 DNA has been used as a hybridization probe to detect nuclear RNA, and Cot-1 RNA strictly localizes to the interphase chromosome territories in *cis* and remains stably associated with the chromosome territory following prolonged transcriptional inhibition [53]. Cot-1 RNA localization to chromosome territories is stable following transcription and remains associated with the parent chromosome in euchromatic regions [53]. Cot-1 RNA associates with known nuclear scaffold proteins, including scaffold attachment factor A (SAF-A), and the distribution of these components inversely correlates with chromatin compaction in normal and experimentally manipulated nuclei, suggesting that Cot-1 RNA plays a fundamental role in chromosome biology, promoting an open chromatin state, and may act as a bridge to non-chromatin elements within the nucleus [53]. All three known ASAR lncRNAs (and the XIST lncRNA) contain a high concentration of L1 elements (>30% of the expressed sequence; [40,41,45]), and in recent experiments it was found that human Cot-1 DNA can detect ASAR6 RNA when expressed from a BAC transgene integrated into mouse chromosomes (unpublished observations). It has been proposed that all human autosomes express ASAR lncRNAs [41], and taken with the observation that the Cot-1 RNA FISH signal is predominantly due to L1 sequences [53], and is detected on all autosomes, suggests that at least some of the RNA FISH signal detected by Cot-1 represents ASAR lncRNAs expressed from every autosome.

## 5. Allele-Specific Mechanisms of Action of lncRNAs

lncRNAs may act through transcript- and transcription-dependent mechanisms, including activation or repression of neighboring genes, mediating chromosomal interactions, forming nuclear structures, forming R-loops, acting as a guide or decoy for transcription factors, sponging microRNAs, regulating post-transcriptional decay of mRNAs, or regulating the cellular localization of DNA or RNA binding proteins [54]. XIST has long served as a paradigm to study lncRNA mechanism, whereby function with associated proteins is thought to mediate outcomes. XIST RNA interacts with at least 81 proteins from the chromatin modification, nuclear matrix, and RNA remodeling pathways, including hnRNP K, required for chromatin modifications and Polycomb group recruitment, and SPEN, which is required for gene silencing [55]. The matrix attachment protein SAF-A is required for normal localization of Xist onto chromatin, and may be particularly relevant during XCI maintenance [56]. The loss of zinc-finger DNA binding protein CIZ1 leads to a diffuse nuclear distribution of XIST RNA, and the loss of SPEN leads to a defect in long range localization of Xist RNA and reduced Xist stability [32].

An exciting direction of research has emerged that may explain complex lncRNA functions, whereby lncRNAs act as architectural scaffolding molecules that promote liquid-liquid phase separated (LLPS) superstructures (also termed membraneless organelles) with diverse RNAs and proteins [57]. Collectively known as the liquid nucleome, LLPS allows for vast subcompartmentalization of the nucleus, of which the potential functional outcome may encompass protein, RNA, and DNA interactions [58]. On XIST RNA, the E-repeat domain facilities high density binding of RNA-binding proteins PTBP1, TDP-43, and CELF1, to create a physical condensate, which is critical for the maintenance of gene silencing on the inactive X [59]. Polycomb group proteins play a major role in X inactivation and gene silencing, and have also been observed forming phased separated condensates [60], further strengthening the role of lncRNAs in subnuclear structure and widespread gene regulation. The lncRNA NEAT1 acts as a scaffold inside the paraspeckle, a dynamic nuclear body capable of sequestering RNA-binding proteins, miRNAs and mRNAs, and affecting the DNA damage response pathway [61,62].

An interesting clue to the function of some lncRNAs is their high density of LINE1(L1) retrotransposon derived sequence, as observed in the Cot-1 RNAs, ASARs, Firre, and Xist [37,49]. L1s have been long hypothesized as “booster elements” that function during the spreading of X-chromosome inactivation, and have a low concentration in regions of the X chromosome that escape inactivation [63]. L1s are implicated in mono-allelic gene expression on autosomes due to their high density at both imprinted and RME loci [64]. Antisense L1s are recruiters of RNA-binding proteins, and when present within lncRNAs may provide a means of achieving protein-mediated effects [65]. Antisense L1 RNA is also known to suppress splicing at cryptic splice sites and 3′ end processing, functions that support the non-canonical structure of the unspliced ASAR, vlinc- or macro- RNAs [65].

Lastly, allele-specific chromatin structure differences are now beginning to be uncovered through the advent of inexpensive deep-sequencing assays and the achievement of accessible haplotype resolved model systems [66,67]. Mono-allelic lncRNAs have evolved mechanisms for homolog-specific regulation in *cis* and in *trans*, and their role in allele-specific chromosome behavior is worthy of further study.

## 6. Conclusions

lncRNAs make up a significant portion of the mammalian transcriptome, show tissue-specific expression, and unlike coding genes have greatly increased in number during recent evolution, making them an interesting candidate to explain recent development of highly complex regulatory units and organ structures. Of the few lncRNAs that have been well studied to date, several notable examples display DAE by either genomic imprinting or RME mechanisms. A general explanation of how and why lncRNAs are well suited to regulate DAE has not yet been found. Additionally, how allele-specific lncRNAs can remain associated with chromosomes in *cis*, or dissociate and function specifically in *trans* on the distant opposite homolog is a yet unexplored area. In conclusion, imprinted and RME lncRNAs comprise special regulatory units critical to multiple biological processes, and the further study of these regions and lncRNA effectors is likely to yield yet unknown fundamental behavior of the genome.

## Figures and Tables

**Figure 1 ncrna-07-00066-f001:**
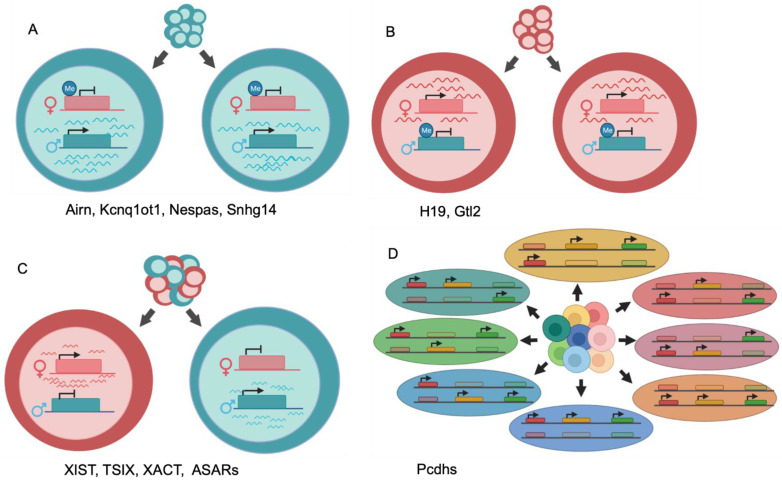
Generalized illustrations of differential allelic expression. (**A**) Maternal imprinting and paternal-specific expression of a single locus for all cells within a population. (**B**) Paternal imprinting and maternal-specific expression of a single locus for all cells within a population. (**C**) Random monoallelic expression of a single locus yields a population with mixed paternal and maternal expression. (**D**) Random monoallelic expression of multiple loci across a chromosome generates diversity of allelic combinations. Three expressed loci on one chromosome pair produces eight possible allelic combinations.

## Data Availability

Not applicable.

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
