# Peer review of "Differential Allelic Expression among Long Non-Coding RNAs"

_ncrna, 2021, doi:10.3390/ncrna7040066_

Round 1

Reviewer 1 Report

‘Differential alleles expression of long non-coding RNAs’ by Heskett, Spellmen, and Thayer is a nice review of the subset of lncRNAs that are monoallelically expressed and I only have relatively minor comments. First, I find the title is a little misleading because it implies that lncRNAs tend to be monoallelically expressed. I suggest changing it to ‘Long non-coding RNAs that are monoallelically expressed’ or ‘Long non-coding RNAs whose alleles are differentially expressed’. There is a very provocative statement early in the review: “it has been conservatively estimated that at least 62% of the genome is consistently transcribed.” Are the authors saying that 62% of the genome is transcribed in all cells? I find that hard to believe and they don’t reference a paper showing that. Thinking about it, one could say global genes are ‘consistently transcribed’ and refer to these genes being transcribed in erythrocytes. If this is what they mean, then they should state it more clearly because it currently reads like “62% of the genome is transcribed in all cells”. Xist-deficiency also has been reported to cause DRT but this wasn’t mentioned. One of the most interesting things about chromosomal instability that results when ASARs or Xist is deleted is that it may explain the chromothripsis detected in cancer cells. I recommend at least briefly discussing chromothripsis in this review. On Page 6, “Utilizing genetic deletion or ectopic integration assays of a full-length L1PA2 transposable element (oriented in the antisense direction with respect to the L1PA2)” Is unclear. How can a full-length L1PA2 transposable element be oriented in the antisense direction with respect to the L1PA2?

Author Response

We thank the reviewer for fair and helpful comments. We accept the critiques from Reviewer 1 and have made the following changes:

  1. We have changed the preposition in the title to "among", to avoid implying that  lncRNAs tend to be monoallelically expressed
  2. We have clarified the statement about 62% of the genome being transcribed. This is directly from the results of the ENCODE 2012 paper in Nature, and we have more accurately paraphrased the ENCODE authors in stating that when studying multiple tissue types and multiple subcellular fractions, 62% of genomic bases are reproducibly covered in the RNA-sequencing data. 
  3. We have cited a major publication and stated that DRT has been observed on the X chromosome after Xist disruption. This is an important point to show that disruption of ASARs or Xist lncRNAs leads to a similar outcome on their respective chromosomes.
  4. We have discussed chromothripsis, and suggested that delayed mitotic condensation as a result of ASAR disruption could potentially be a precursor of a chromothriptic event.
  5.  We have clarified the statement about the sense orientation of the L1PA2 element.

Reviewer 2 Report

The review article authored by Heskett et al., has done a good job of describing the Differential allelic expression of long non-coding RNAs. The article highlights the current understanding of lncRNAs expression from one chromosome homolog which is also known as mono-allelic, or Differential-Allelic Expression.

Differential allelic expression of lncRNAs by genomic imprinting and its role in X-inactivation were well put together and presented with clear examples.

Graphical representations were nicely done.

Minor comment:

Kindly correct the typographical error in line 118 – “Shng14” to “Snhg14”

Author Response

We thank the reviewer for their feedback and have addressed the minor comment in the main text.

Reviewer 3 Report

In this review, Heskett et al. present a quick overview of the important fact that many long non-coding RNAs (lncRNAs) display differential allelic expression (DAE), and present several examples. Some of these examples are well-known (Xist) while others are less commonly mentioned (Kcnq1ot1), and therefore this review is quite interesting in insisting on this important aspect of lncRNA biology.

The field of lncRNA biology has shown that it can be quite difficult to distinguish between key lncRNA, which function as master regulators in the cell, and others which are more likely due to pervasive transcription. We agree with the authors’ premise, which is that lncRNAs showing DAE deserve special scrutiny, e.g. for the fact that this DAE can often provide a molecular mechanism for cis interference effects.

Overall, the review is well-written and quite clear, and provides a good background on the importance and examples of monoallelism. It provides a clear reminder that many well-known lncRNAs also display monoallelism, which is not always mentioned in reviews on lncRNAs. In that regard, the mention of the enigmatic ASAR lncRNAs, and of adjacent protein-coding genes displaying DAE, was particularly interesting.

The archetypal example of lncRNA DAE is Xist, which is expressed from the inactive X and causes its silencing in cis. Due to the nature of X-chromosome silencing, other X-linked lncRNAs unsurprisingly display DAE (Tsix, Firre, XACT), which is nevertheless important to mention.

The POE studies would gain from being expanded upon (lines 122-125). In the cited study (ref 28), four out of nine significant opposite POE effects corresponded to lncRNAs, which could be mentioned: LINC00607, MIR4444-1, LINC01055 and potentially MIRR4454.

The main weakness of this review is that while Figure 1 is simple yet informative, Figure 2 is very simplistic and does not add much to the review—which is unfortunate because it does not match the quality of the text. I would strongly suggest strengthening this Figure, maybe by selecting specific lncRNA examples, with annotation, rather than a “blank” template summarizing the now well-known facts that lncRNAs can result in transcriptional interference, mediate RNP formation and/or chromatin. Also, the notation of panels 2B and 2C is usually switched in most journals (left-to-right before top-to-bottom).

Furthermore, it is pretty clear that the figures were created using the tool BioRender; as per the license agreement, this should be mentioned, either in the legends of the figure or in the “acknowledgments” section, which is currently not filled:

https://help.biorender.com/en/articles/3619405-how-do-i-cite-biorender

Minor typo:

line 96: “(...) lncRNAs who’s function (...)” -> “(...) lncRNAs whose function (...)”

Author Response

We thank the reviewer for fair and helpful comments. We accept the critiques from Reviewer 3 and have made the following changes:

  1. We have added an additional statement with a specific example of a lncRNA that contains a variant that exhibits parent of origin effects.
  2. We have added the examples discussed in the main text as labels to Figure 1, and have listed the panels in left-to-right order.
  3. We have removed figure 2. With consideration, we have observed that there are many excellent and detailed examples of lncRNA mechanism-of-action figures in the literature, and it is not helpful to include another in this review.
  4. We have cited biorender in the acknowledgements section, as recommended by the biorender software manufacturers.
  5. We have fixed the minor typo on line 96.

Round 2

Reviewer 1 Report

It looks good to me.  It is ready to publish.

Reviewer 3 Report

In this revised version of the manuscript, Heskett et al. have addressed my previous comments, as well as made the decision to remove Figure 2. I agree with the authors that as many reviews in the lncRNA field already present many mechanistic examples and as the originality of this review is to present monoallelism, this decision is justified. The text is well-written and clear, and should be of broad interest to researchers in the field.